# Development and external validation of machine learning algorithms for postnatal gestational age estimation using clinical data and metabolomic markers

Steven Hawken[1]*, Robin Ducharme[1], Malia S. Q. Murphy[1], Brieanne Olibris[1], A. Brianne Bota[1], Lindsay A. Wilson[1], Wei Cheng[1], Julian Little[2], Beth K. Potter[2], Kathryn M. Denize[3], Monica Lamoureux[3], Matthew Henderson[3], Katelyn J. Rittenhouse[4], Joan T. Price[4], Humphrey Mwape[5], Bellington Vwalika[6], Patrick Musonda[7], Jesmin Pervin[8], A. K. Azad Chowdhury[9], Anisur Rahman[8], Pranesh Chakraborty[3], Jeffrey S. A. Stringer[4], Kumanan Wilson[1,2,10]

1 Clinical Epidemiology Program, Ottawa Hospital Research Institute, Ottawa, Ontario, Canada, 2 School of Epidemiology and Public Health, University of Ottawa, Ottawa, Ontario, Canada, 3 Newborn Screening Ontario, Children's Hospital of Eastern Ontario, Ottawa, Ontario, Canada, 4 University of North Carolina at Chapel Hill, Chapel Hill, North Carolina, United States of America, 5 UNC Global Projects Zambia, Lusaka, Zambia, 6 Department of Obstetrics and Gynaecology, University of Zambia School of Medicine, Lusaka, Zambia, 7 Department of Medical Statistics, University of Zambia College of Public Health, Lusaka, Zambia, 8 International Centre for Diarrhoeal Disease Research, Dhaka, Bangladesh, 9 Dhaka Shishu (Children) Hospital, Dhaka, Bangladesh, 10 Faculty of Medicine, Department of Medicine, University of Ottawa, Ottawa, Ontario, Canada

* shawken@ohri.ca

**Data Availability Statement:** We have uploaded the study datasets to Dryad. The link is as follows: https://doi.org/10.5061/dryad.m37pvmd6b.

## Abstract

### Background

Accurate estimates of gestational age (GA) at birth are important for preterm birth surveillance but can be challenging to obtain in low income countries. Our objective was to develop machine learning models to accurately estimate GA shortly after birth using clinical and metabolomic data.

### Methods

We derived three GA estimation models using ELASTIC NET multivariable linear regression using metabolomic markers from heel-prick blood samples and clinical data from a retrospective cohort of newborns from Ontario, Canada. We conducted internal model validation in an independent cohort of Ontario newborns, and external validation in heel prick and cord blood sample data collected from newborns from prospective birth cohorts in Lusaka, Zambia and Matlab, Bangladesh. Model performance was measured by comparing model-derived estimates of GA to reference estimates from early pregnancy ultrasound.

### Results

Samples were collected from 311 newborns from Zambia and 1176 from Bangladesh. The best-performing model accurately estimated GA within about 6 days of ultrasound estimates

**Funding:** This study was supported by the Bill & Melinda Gates Foundation [OPP1141535] (KW), [OPP1033514] (JSAS); with site support to Zambia provided by the US National Institutes of Health [P30 AI50410, D43 TW009340]. The funders had no role in study design, data collection and interpretation, or the decision to submit the work for publication.

**Competing interests:** The authors have declared that no competing interests exist.

in both cohorts when applied to heel prick data (MAE 0.79 weeks (95% CI 0.69, 0.90) for Zambia; 0.81 weeks (0.75, 0.86) for Bangladesh), and within about 7 days when applied to cord blood data (1.02 weeks (0.90, 1.15) for Zambia; 0.95 weeks (0.90, 0.99) for Bangladesh).

## Conclusions

Algorithms developed in Canada provided accurate estimates of GA when applied to external cohorts from Zambia and Bangladesh. Model performance was superior in heel prick data as compared to cord blood data.

## Introduction

Preterm birth is a leading cause of neonatal morbidity and mortality worldwide that disproportionately affects infants born in low- and middle-income countries (LMIC) [1]. Knowledge of the burden of preterm birth across LMIC is essential to evaluating the impact of policies and programs aimed at improving pregnancy and neonatal outcomes. Accurate estimates of preterm birth rates rely on the use of standard definitions across jurisdictions and consistent reporting of pregnancy outcomes. International preterm birth surveillance efforts are hampered in many LMIC by limited access to ultrasound dating technology and poor recall reliability of a woman's last menstrual period [2–4]. The reliability of commonly used newborn assessments for estimating gestational age (GA) postnatally is often limited in preterm and growth-restricted infants, and these methods are also subject to high inter-user variability, making estimation of the burden of preterm birth in many jurisdictions problematic [5–9].

There is now a push by health organizations for strengthened data surveillance systems and novel tools that can more accurately assess and monitor rates of preterm birth in low resource countries, many having some of the highest preterm birth rates globally [10, 11]. Our research team has previously investigated the distribution of analyte levels obtained from newborn screening data in Ontario infants according to GA at birth and found that many analytes included in the newborn screening panel varied with level of prematurity, including a number of amino acids, acylcarnitines and fatty acid oxidation markers, 17-OHP, thyroid stimulating hormone and the relative proportions of fetal and adult hemoglobin subtypes [12, 13]. Based on these findings, we developed algorithms in a North American cohort of newborns to estimate GA postnatally using newborn screening analyte levels from newborn dried blood spots and a small number of clinical covariates. We have demonstrated that these previously developed models provided accurate estimates to within about 1–2 weeks of ultrasound-based GA [14–17]. Previous models were developed using conventional multivariable linear and logistic regression methods and have been internally validated in specific ethnic subgroups of the Canadian population [18], and externally validated in a cohort of infants born in Matlab, Bangladesh [19], demonstrating satisfactory performance, but lower accuracy of GA predictions compared to the setting in which models were developed. In recent years, a number of powerful machine learning (ML) methods have been developed that allow the efficient handling of large databases, large numbers of predictors, and the incorporation of non-linear associations and complex interactions. We sought to redevelop our models by incorporating advanced ML methods with the aim of dramatically improving model performance. In this study, we report the performance of our newly developed ML-based GA estimation algorithms using ELASTIC NET regression [20] in two international prospective birth cohorts from Lusaka, Zambia, and Matlab Bangladesh.

## Methods

### Ethics approval and consent to participate

Written informed consent was obtained from all participants from Zambia and Bangladesh before any data or samples were collected. Approval for the Zambia cohort was obtained from the University of Zambia School of Medicine Biomedical Research Ethics Committee (Reference number: 016-04-14), and the University of North Carolina School of Medicine (Study number: 14–2113). Approval from the Bangladesh cohort was obtained from the Research Review and Ethical Review Committees of the International Centre for Diarrhoeal Disease Research, Bangladesh (PR-16039). Approval was also obtained from the Ottawa Health Science Network Research Ethics Board (20160219-01H) and the Children's Hospital of Eastern Ontario Research Ethics Board (16/20E) for model development in the Ontario, Canada cohort, and external validation in the Zambia and Bangladesh cohorts.

### Study design and research participants

We sought to evaluate the performance of a GA estimation algorithm developed in a retrospective population cohort of infants from Ontario, Canada in prospective birth cohorts of infants born in Lusaka, Zambia and Matlab, Bangladesh. Detailed descriptions of all three cohorts have been published previously [17, 21, 22]. A summary of the cohorts is available in Table 1.

### Sample collection and analysis

Details of sample collection and analysis are included in the Supplemental Methods in S1 File. All models were developed and internally validated based on clinical covariates and laboratory results from heel prick blood samples in the Ontario cohort. For the Zambia and Bangladesh cohorts, a subset of infants had only a heel prick or a cord blood sample collected, and a further subset had both sample types. All samples from the Ontario, Zambia and Bangladesh cohorts were analyzed centrally at the Newborn Screening Ontario (NSO) laboratory in Ottawa, Canada. Analytes were included as candidate predictors in GA estimation models based on their routine measurement as part of Ontario's expanded newborn screening program, including hemoglobin profiles, amino acids, acylcarnitines, hormone and endocrine markers, enzymes and co-enzymes (Table 2). Management of incidental clinical findings (screen-positive cases) for conditions screened for by the NSO program identified in the course of analyzing international samples have been reported elsewhere [23, 24].

### Statistical analysis

All analyses were conducted using SAS 9.4 and R 3.3.2. Additional details of statistical methods are provided in the Supplemental Methods in S1 File. Data preparation steps, including standardization and log transformation were applied uniformly in the Ontario, Zambia and Bangladesh cohorts. A complete case analysis was used for model derivation and internal

**Table 1. Summary of Ontario, Zambia and Bangladesh cohorts.**

| Cohort | Site | Cohort Type | Enrollment | GA Estimate |
|---|---|---|---|---|
| **Ontario, Canada** | All hospital births and midwife attended home births | Retrospective birth registry | Infants born between January 2012 and December 2014 | First trimester ultrasound |
| **Lusaka, Zambia** | Women and Newborn Hospital of the University Teaching Hospital | Nested prospective within the Zambian Preterm Birth Prevention Study (ZAPPS) | Women <20 weeks gestation between August 2015 and September 2017 | Ultrasound at first prenatal visit |
| **Matlab, Bangladesh** | International Centre for Diarrhoeal Disease Research, Bangladesh | Nested prospective within the Preterm and Stillbirth Study, Matlab (PreSSMat) | Women 11–14 weeks gestation between January 2017 and July 2018 | Ultrasound at enrollment |

**Table 2. Newborn screening analytes included in predictive models.**

| Hemoglobins | Adult hemoglobin: HbA(A) |
|---|---|
| | Fetal hemoglobin: HbF (F), Acetylated HbF (F1) |
| **Endocrine markers** | 17-hydroxyprogesterone (17-OHP), Thyroid stimulating hormone (TSH) |
| **Amino Acids** | Arginine (arg); phenylalanine (phe); alanine (ala); leucine (leu); ornithine (orn); citruline (cit); tyrosine (tyr); glycine (gly); methionine (met); valine (val); |
| **Acyl-carnitines** | C0; C2; C3; C4; C5; C5:1; C6; C8; C8:1; C10; C10:1; C12; C12:1; C14; C14:1; C14:2; C16; C18; C18:1; C18:2; C10:1; C12:1; C14:1; C14:2; C4OH; C5:1; C5DC; C5OH; C6DC; C16:OH; C16:1OH; C18OH; C18:1OH; C3DC; C4DC |
| **Enzyme markers** | Biotinidase; immunotripsinogen |

validation in Ontario due to very low missingness and the large available sample size. A small proportion of subjects in the Zambia and Bangladesh cohorts had missing values for a small subset of analytes (due to sample quality degradation or insufficient blood spot volume) and these values were multiply imputed (additional details in Supplemental Methods in S1 File).

Three distinct GA estimation models were derived. Models were fit and all parameters estimated using both the Ontario training subset (N = 79,636) and validation subset (N = 39,829) of the Ontario cohort:

**Model 1:** *Baseline model* containing only infant sex, multiple birth (yes/no), birth weight (grams), and pairwise interactions among these covariates.

**Model 2:** *Analytes model* including infant sex, multiple birth (yes/no), newborn screening analytes (listed in Table 2), and pairwise interactions among covariates.

**Model 3:** *Full model* containing infant sex, multiple birth (yes/no), birth weight (grams), newborn screening analytes, and pairwise interactions among these covariates.

The analytes most strongly associated with GA were identified using partial spearman correlation analysis, and these, in addition to birthweight, were modeled using restricted cubic splines with 5 knots to allow for non-linearity. All other analytes were modeled with linear terms. To address the large imbalance in number of term infants compared to preterm and post-term infants, we applied a weighting scheme in the regression model fitting process to balance the influence of term infants with less common gestational ages (additional details in Supplemental Methods in S1 File).

Given the large number of covariates and interactions included, Model 2 and Model 3 were fit using an ELASTIC NET ML approach [20].

Final Ontario model equations were used to calculate an estimated GA in the test subset of the Ontario cohort that had no role in model development, as well as in the Zambia and Bangladesh external validation cohorts. For each infant, model performance was assessed by comparing the estimated GA from the model to the ultrasound-derived GA and calculating the mean absolute error (MAE) measured in weeks (the average of the absolute difference between model-based vs. ultrasound-based values across all observations). Lower MAE reflects more accurate model-based GA estimates. We also calculated the percentage of infants with GAs correctly estimated within 7 days of ultrasound-based GA. We assessed model performance overall and in important subgroups: preterm birth (<37 weeks gestation), and small-for-gestational age: below the 10th (SGA10) and 3rd (SGA3) percentile for birth weight within categories of gestational week at delivery and infant sex, based on INTERGROWTH-21 categories [25]. 95% bootstrap percentile confidence intervals were calculated for all validation performance metrics, which also took account of data imputation in the Bangladesh and Zambia cohorts.

### Estimated preterm birth rate

We estimated the preterm birth rate by calculating the proportion of model-based GA estimates that were below 37 weeks, as well as 95% bootstrap confidence intervals, and compared these to the observed preterm birth rate in each cohort (based on ultrasound GA).

### Classification accuracy

Although our models were intended to estimate gestational age in weeks on a continuous scale and not intended or optimized for classification, we calculated the accuracy, sensitivity and specificity of continuous Model 3 estimates when used to classify infants as term or preterm in heel prick data for Ontario, Bangladesh and Zambia. Depending on the intended application, a logistic regression or other methodology better suited to prediction of a dichotomous outcome would be preferred, and cut-points could be adjusted to prioritize sensitivity or specificity in terms of classifying infants.

## Results

### Participant characteristics

The Ontario internal validation (testing subset) cohort included 39,666 infants (Fig 1). In the Ontario cohort, the proportion of infants born preterm (GA<37 weeks) based on ultrasound was 5.6%, and 3.9% and 0.92% of infants were classified as SGA10 and SGA3, respectively. In the external validation cohorts, a total of 142 heel prick samples and 265 cord blood samples collected from 311 unique newborns from Zambia, and 520 heel prick samples and 1139 cord blood samples collected from 1176 unique newborns from Bangladesh. 32 (22.5%) heel and 99 (37.4%) cord samples from Zambia were missing one or more analytes, with a maximum of three missing in any sample. 28 (5.4%) heel and 22 (1.9%) cord samples from Bangladesh were missing one or more analytes, with a maximum of 5 and 3 missing values respectively (one subject missing all analyte values was removed from the analysis). Based on ultrasound GA, the preterm birth proportions were similar in the Zambia and Bangladesh cohorts (9.6% versus 9.7%). The Zambia cohort had a much lower proportion of newborns classified as SGA10 as

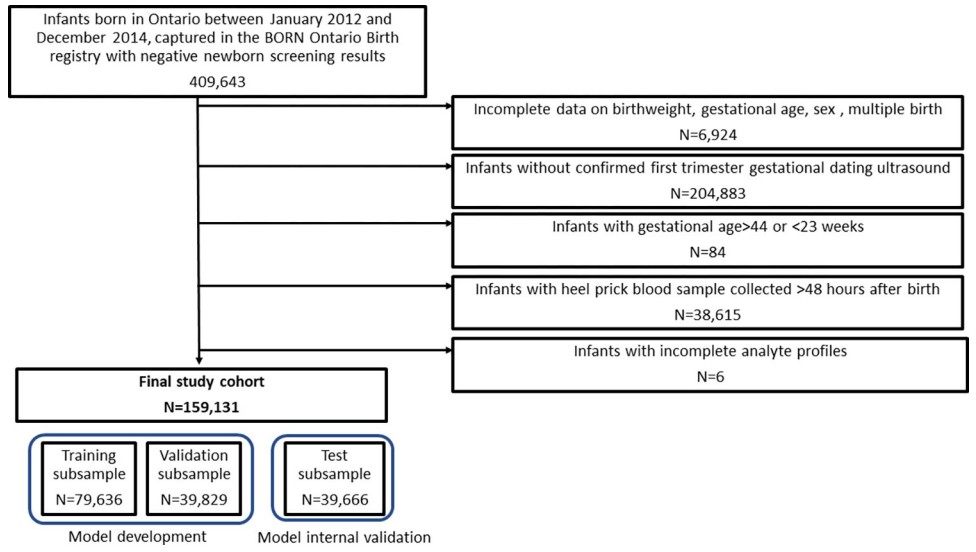

**Fig 1. Cohort creation.**

**Table 3. Infant characteristics.**

| | Ontario | Zambia | | | Bangladesh | | |
|---|---|---|---|---|---|---|---|
| | Unique Infants Overall (n = 39,666) | Heel Prick Samples (n = 142) | Cord Blood Samples (n = 265) | Unique Infants Overall (n = 311) | Heel Prick Samples (n = 520) | Cord Blood Samples (n = 1139) | Unique Infants Overall (n = 1176) |
| Sex, n (%) | | | | | | | |
| Male | 19536 (49.3%) | 66 (46.5%) | 127 (47.9%) | 151 (48.5%) | 258 (49.6%) | 577 (50.7%) | 593 (50.4%) |
| Female | 20130 (50.7%) | 76 (53.5%) | 138 (52.1%) | 160 (51.5%) | 262 (50.4%) | 562 (49.3%) | 583 (49.6%) |
| Gestational Age (wks), mean (SD) | 39.3 (1.57) | 38.7 (1.64) | 38.6 (2.09) | 38.5(2.10) | 38.8 (1.5) | 38.7 (1.7) | 38.6 (1.7) |
| Gestational Age (wks), n (%) | | | | | | | |
| $\geq$37 weeks | 37440 (94.4%) | 131 (92.3%) | 243 (91.7%) | 281 (90.4%) | 485(93.3%) | 1031 (90.5%) | 1063 (90.4%) |
| $32^\circ-36^6$ weeks | 2049 (5.2%) | 11 (7.7%) | 18 (6.8%) | 26 (8.4%) | 34 (5.5%) | 105 (9.2%) | 109 (9.3%) |
| $28^\circ-31^6$ weeks | 126 (0.3%) | 0 (0%) | 2 (0.8%) | 2 (0.6%) | 1 (0.2%) | 2 (0.2%) | 3 (0.3%) |
| <28 weeks | 51 (0.1%) | 0 (0%) | 2 (0.8%) | 2 (0.6%) | 0 (0.0%) | 1 (0.1%) | 1 (0.1%) |
| Birth Weight (g), mean (SD) | | | | | | | |
| Overall | 3379 (530) | 3128 (485) | 3102 (529) | 3086 (545) | 2835 (429) | 2860 (440) | 2855 (445) |
| Term infants | 3431 (476) | 3185 (429) | 3186 (419) | 3179 (436) | 2876 (390) | 2911 (397) | 2910 (396) |
| Preterm infants | 2504 (623) | 2456 (620) | 2178 (719) | 2224 (699) | 2267 (546) | 2369 (522) | 2340 (544) |
| Small for Gestational Age, n (%) | | | | | | | |
| SGA10 | 1561 (3.9%) | 11 (7.8%) | 30 (9.6%) | 30 (9.7%) | 147 (28.3%) | 286 (25.1%) | 297 (25.3%) |
| SGA3 | 363 (0.9%) | 3 (2.1%) | 11 (3.5%) | 14 (4.5%) | 61 (11.7%) | 107 (9.4%) | 112 (9.5%) |
| Multiple Birth, n (%) | 1069 (2.7%) | 8 (5.6%) | 14 (4.5%) | 14 (4.5%) | 7 (1.4%) | 19 (1.7%) | 19 (1.6%) |

compared to the Bangladesh cohort (9.7% vs. 25.3%). Characteristics of unique infants in each cohort, as well as the infants represented in heel prick and cord blood sample cohorts are provided in Table 3.

## Model performance using heel prick data from Ontario, Zambia and Bangladesh

Model external validation performance was largely similar between the Zambia and Bangladesh cohorts. Accuracy of estimated GA was generally lower in the external validation cohorts than in the Ontario internal validation cohort for the same models. In general, Model 3, which included both clinical and analyte predictors, outperformed Models 1 and 2. Accuracy of GA estimates in all models was highest in term infants and tended to be lower in preterm and SGA infants, but Model 3 consistently provided the most accurate estimates across the spectrum of GA (Table 4 and Fig 2). Model 1 estimated GA to within approximately 7 days of ultrasound-assigned GA, with similar performance among samples from Ontario, Zambia and Bangladesh (MAE (95% CI) 0.96 (0.95,0.97), 0.96 (0.82,1.12) and 0.99 (0.92,1.06) weeks, respectively). Model 3 estimated GA to within 5 days of ultrasound-assigned values when applied to the Ontario data (MAE (95% CI) 0.71 (0.71, 0.72) and within 6 days in the Zambia and Bangladesh data (MAE (95% CI) 0.79 (0.69, 0.90) and 0.81 (0.75, 0.86), respectively). GA was correctly estimated to within 1 week of ultrasound-assigned values for 74.6% (95% CI 74.2,75.1), 69.4% (60.6, 77.5) and 68.4% (64.3, 72.4) of heel prick samples from Ontario, Zambia and Bangladesh, respectively.

**Table 4. Heel prick samples: MAE and proportion estimated within 1 week of ultrasound-assigned gestational age.**

| | Ontario | | | | Zambia | | | | Bangladesh | | | |
|---|---|---|---|---|---|---|---|---|---|---|---|---|
| | Overall, N = 39,666 | <37 wks N = 2226 | SGA10, N = 1561 | SGA3, N = 363 | Overall, N = 142 | <37 wks N = 11 | SGA10, N = 11 | SGA3, N = 3 | Overall, N = 520 | <37 wks, N = 35 | SGA10, N = 147 | SGA3, N = 61 |
| **Model 1: Baseline Model** | | | | | | | | | | | | |
| MAE (95% CI) | **0.96 (0.95, 0.97)** | 1.76 (1.71, 1.81) | 2.70 (2.65, 2.75) | 3.85 (3.76, 3.95) | **0.96 (0.82, 1.12)** | 2.12 (1.1, 3.4) | 1.79 (1.09, 2.68) | 3.54 (2.42, 5.38) | **0.99 (0.92, 1.06)** | 2.15 (1.73, 2.57) | 1.14 (1.00, 1.28) | 1.73 (1.49, 1.97) 26.1 (14.8, 37.7) |
| % +/-1 wk (95% CI) | **60.4 (59.9, 60.9)** | 31.0 (29.2, 32.8) | 0.45 (0.13, 0.82) | 0.55 (0.00, 1.40) | **63.3 (55.3, 71.1)** | 27.3 (0.0, 57.1) | 36.6 (10.0, 66.7) | 0.0 (0.0, 0.0) | **61.1 (56.9, 65.3)** | 28.3 (14.3, 44.8) | 55.7 (47.8, 63.7) | |
| **Model 2: Analyte Model** | | | | | | | | | | | | |
| MAE (95% CI) | **0.79 (0.79, 0.80)** | 1.25 (1.21, 1.29) | 0.90 (0.86, 0.94) | 1.03 (0.92, 1.13) | **0.95 (0.81, 1.12)** | 1.70 (1.15, 2.32) | 0.86 (0.53, 1.29) | 1.09 (0.48, 2.10) | **0.92 (0.86, 0.99)** | 1.8 (1.5, 2.1) | 0.96 (0.85, 1.08) | 0.99 (0.83, 1.16) 60.3 (47.4, 72.3) |
| % +/-1 wk (95% CI) | **69.4 (69.0, 69.9)** | 46.9 (44.9, 48.9) | 66.6 (64.2, 68.9) | 62.0 (56.6, 66.9) | **64.9 (55.6, 73.2)** | 27.5 (0.0, 56.4) | 73.4 (41.7, 100) | 73.6 (0.0, 100.0) | **64.9 (60.6, 69.2)** | 22.7 (9.1, 37.8) | 59.4, (51.0, 67.7) | |
| **Model 3: Full Model** | | | | | | | | | | | | |
| MAE (95% CI) | **0.71 (0.71, 0.72)** | 1.03 (0.99, 1.06) | 1.13 (1.09, 1.17) | 1.48 (1.37, 1.60) | **0.79 (0.69, 0.90)** | 1.49 (1.07, 1.93) | 0.87 (0.46, 1.32) | 1.29 (0.12, 2.17) | **0.81 (0.75, 0.86)** | 1.44 (1.16, 1.70) | 0.81 (0.72, 0.91) | 0.97 (0.81, 1.15) 58.5 (45.5, 70.8) |
| % +/-1 wk (95% CI) | **74.6 (74.2, 75.1)** | 58.0 (55.9, 60.2) | 50.5 (48.2, 53.0) | 35.8 (30.6, 40.6) | **69.4 (60.6, 77.5)** | 28.3 (0.0, 60.0) | 60.1 (25.0, 90.9) | 29.8 (0.0, 100.0) | **68.4 (64.3, 72.4)** | 32.1 (17.2, 48.6) | 66.0 (58.2, 73.6) | |

Data are presented as the mean and 2.5th and 97.5th bootstrap percentiles for MAE, and the percentage of model estimates within 1 week of ultrasound GA for 2000 bootstrap samples in each cohort.

When applied to samples from preterm infants in Ontario, Model 3 correctly estimated GA to within 7 days of ultrasound-assigned values (MAE (95%CI) 1.03 (0.99, 1.06)), and performed significantly better than Models 1 and 2 (MAE (95%CI) 1.76 (1.71, 1.81) and 1.25 (1.21, 1.29), respectively). The number of preterm infants in the external validation cohorts was small, with only 11 heel prick samples from Zambia and 35 from Bangladesh, however in both settings, model 3 outperformed both models 1 and 2, and was accurate to within about 10 days on average in preterm infants (MAE (95% CI) 1.49 (1.07, 1.93) and 1.44 (1.16, 1.70), respectively in Zambia and Bangladesh).

When applied to Ontario heel prick samples from SGA infants, Model 2 had the best performance, estimating GA to within about 6 days of ultrasound-assigned values for SGA10 and 7 days for SGA3 subgroups (MAE (95% CI) 0.90 (0.86, 0.94) and 1.03 (0.92, 1.13), respectively). Our validation cohort from Zambia contained only 11 SGA10 and 3 SGA3 classified infants. When applied to these samples, Models 2 and 3 both outperformed Model 1, and estimated GA to within about 6 days of ultrasound-assigned values in SGA10 infants, and within about 9 days in SGA3 infants. Our heel prick sample cohort from Bangladesh contained a larger number of SGA samples (147 SGA10 and 61 SGA3 samples). When applied to these samples from Bangladesh, Model 3 estimated GA to within about 6 days of ultrasound-assigned values in the SGA10 subgroup and within about 7 days in the SGA3 subgroup (MAE (95% CI) 0.81 (0.72, 0.91) and 0.97 (0.81, 1.15), respectively).

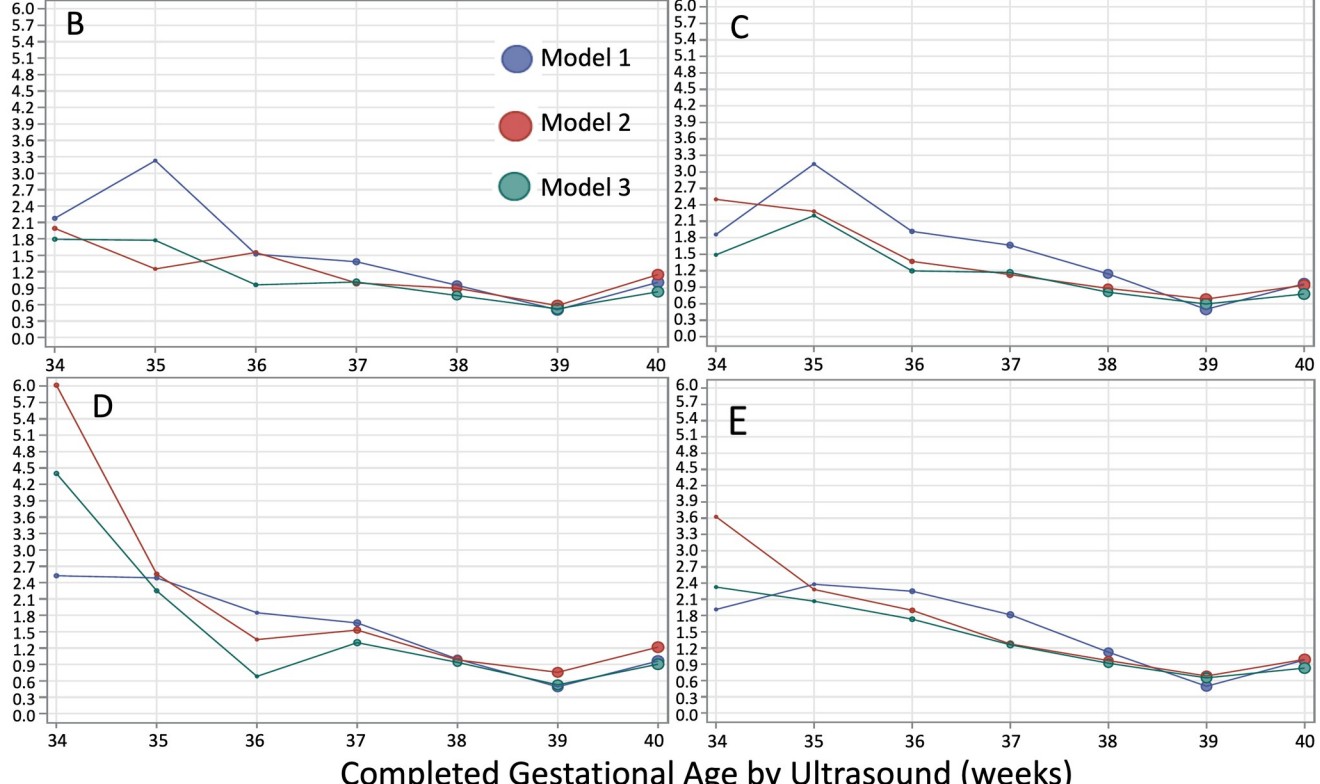

| A | Heel Prick Samples | | Cord Blood Samples | |
|---|---|---|---|---|
| | **Average MAE (weeks)** | | **Average MAE (weeks)** | |
| | Zambia, n=142 | Bangladesh, n=520 | Zambia, n=265 | Bangladesh, n=1139 |
| Model 1: Baseline Model | 0.96 (0.82, 1.12) | 0.99 (0.92,1.06) | 1.03 (0.93, 1.13) | 1.08 (1.03,1.13) |
| Model 2: Analyte Model | 0.95 (0.81, 1.12) | 0.92 (0.86, 0.99) | 1.30 (1.15, 1.47) | 1.06 (1.01,1.11) |
| Model 3: Full Model | 0.79 (0.69, 0.90) | 0.81 (0.75, 0.86) | 1.02 (0.90, 1.15) | 0.95 (0.90, 0.99) |

**Fig 2. Agreement between algorithmic gestational age estimations compared to ultrasound-assigned gestational age.** (A) Legend, and overall MAE (95% CI) for heel prick sample and cord blood samples from Zambia and Bangladesh across three models. Dot size in plots is proportional to sample size in each gestational age category. Performance of each model by ultrasound-validated gestational age when applied to heel prick data in (B) Zambia (C) Bangladesh, and cord blood data in (D) Zambia (E) Bangladesh. MAE, mean absolute error (average absolute deviation of observed vs. predicted gestational age in weeks).

Scatter plots of observed GA versus estimated GA for all three models in the Ontario, Zambia and Bangladesh heel prick cohorts are presented in Fig 3, which shows that in general, lower (preterm) GAs tend to be overestimated by all three models when applied to both external cohorts, with the overestimation being much more pronounced for Model 1 estimates. Similarly, GA tended to be slightly overestimated in post-term infants. These patterns were observed in the Ontario test cohort only for Model 1.

## Model performance using cord blood data from Zambia and Bangladesh

Overall, performance was attenuated when applied to cord blood samples vs. heel prick samples for Models 2 and 3 (the models including analyte covariates), in both the Zambia and Bangladesh cohorts (Table 5 and Fig 2). In the cord blood cohort from Zambia, Models 1 and 3 performed similarly (MAE (95% CI) 1.03 (0.93,1.13) vs. 1.02 (0.90, 1.15)), respectively), and significantly

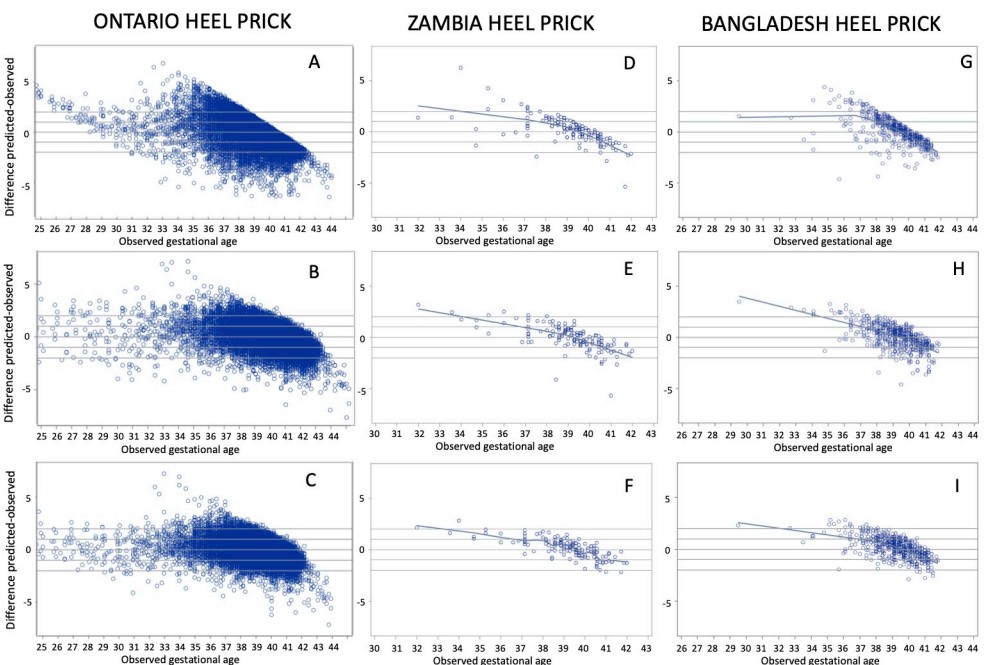

**Fig 3. Residual plots of predicted–observed by ultrasound-assigned gestational age. Heel prick samples from Ontario**: (A) Model 1: Baseline Model, (B) Model 2: Analyte Model, and (C) Model 3: Full Model. **Heel prick samples from Zambia**: (D) Model 1: Baseline Model, (E) Model 2: Analyte Model, and (F) Model 3: Full Model. **Heel prick samples from Bangladesh:** (G) Model 1: Baseline Model, (H) Model 2: Analyte Model, and (I) Model 3: Full Model.

outperformed Model 2 (MAE (95% CI) 1.30 (1.15, 1.47)) (Table 5). Overall, in cord blood samples from Bangladesh, Model 3 significantly outperformed Models 1 and 2 in accurately estimating GA (MAE (95% CI) 0.95 (0.90, 0.99) vs. 1.08 (1.03, 1.13) and 1.06 (1.01, 1.11), respectively).

When applied to cord blood samples from preterm infants in Zambia (n = 22), all models generally performed poorly. The best performing model in this group was Model 1, with an MAE (95% CI) of 2.42 (1.73, 3.12) and estimated GA within 1 week in 36.3% of infants. Models performed slightly better in estimating preterm GA in cord blood samples from Bangladesh

**Table 5. Cord blood samples: MAE and proportion estimated within 1 week of ultrasound-assigned gestational age.**

| | Zambia | | | | Bangladesh | | | |
|---|---|---|---|---|---|---|---|---|
| | Overall, | <37 wks, | SGA10, | SGA3, | Overall, | <37 wks, | SGA10, | SGA3, |
| | N = 265 | N = 22 | N = 25 | N = 8 | N = 1139 | 108 | N = 287 | N = 107 |
| **Model 1: Baseline Model** | | | | | | | | |
| MAE (95% CI) | **1.03 (0.93, 1.13)** | 2.42 (1.73, 3.12) | 1.52 (1.12, 1.96) | 2.27 (1.40, 3.17) | **1.08 (1.03, 1.13)** | 2.21 (1.98, 2.43) | 1.16 (1.05, 1.27) | 1.86 (1.68, 2.06) |
| % +/-1 wk (95% CI) | **60.0 (54.3, 66.0)** | 36.3 (15.8, 57.1) | 40.0 (20.0, 61.1) | 12.5 (0.0, 40.0) | **56.5 (53.6, 59.2)** | 23.2 (15.5, 31.5) | 54.6 (49.0, 60.3) | 21.5 (13.9, 30.0) |
| **Model 2: Analyte Model** | | | | | | | | |
| MAE (95% CI) | **1.30 (1.15, 1.47)** | 3.96 (2.79, 5.34) | 1.76 (1.14, 2.60) | 2.66 (1.33, 5.03) | **1.06 (1.01, 1.11)** | 2.32 (2.07, 2.58) | 1.11 (1.00, 1.22) | 1.17 (0.97, 1.41) |
| % +/-1 wk (95% CI) | **50.2 (43.8, 56.6)** | 10.8 (0.0, 25.0) | 32.5 (13.8, 52.4) | 12.3 (0.0, 40.0) | **56.8 (53.8, 59.8)** | 13.0 (6.9, 19.8) | 51.0 (45.3, 56.7) | 48.7 (38.5, 58.7) |
| **Model 3: Full Model** | | | | | | | | |
| MAE (95% CI) | **1.02 (0.90, 1.15)** | 3.01 (2.15, 3.99) | 1.15 (0.67, 1.82) | 1.41 (0.25, 3.34) | **0.95 (0.90, 0.99)** | 1.92 (1.74, 2.11) | 0.92 (0.84, 1.00) | 1.10 (0.94, 1.27) |
| % +/-1 wk (95% CI) | **60.7 (54.7, 66.8)** | 13.8 (0.0, 31.3) | 54.6 (33.3, 73.9) | 62.9 (25.0, 100.0) | **61.0 (58.1, 63.8)** | 16.8 (9.8, 23.6) | 60.8 (54.9, 66.4) | 51.4 (41.8, 60.8) |

Data are presented as the mean and 2.5th and 97.5th bootstrap percentiles for MAE, and the percentage of model estimates within 1 week of ultrasound GA for 2000 bootstrap samples.

(n = 108). Model 3 had the best performance, with a MAE (95% CI) of 1.92 (0.90, 0.99) and was able to correctly estimate GA to within 1 week in 16.8% of preterm infants.

Our validation cohort from Zambia contained a slightly larger number of SGA cord blood samples (25 SGA10 and 8 SGA3 samples) compared to heel prick samples (11 SGA10 and 3 SGA3). Model 3 had the best performance in this subgroup, performing similarly in SGA10 and SGA3 infants and correctly estimating GA to within 1 week in 54.6 (95% CI 33.3, 73.9) and 62.9% (95% CI25.0, 100.0) of cord samples, respectively. When applied to cord samples from SGA infants in Bangladesh, Model 3 also had the best performance, and correctly estimated GA to within 1 week in 60.8% (95% CI 54.9, 66.4) and 51.4% (95% CI 41.8, 60.8) of cord samples from SGA10 and SGA3 infants, respectively. Fig 4 presents scatter plots for observed versus model-estimated GA for all three models in the Zambia and Bangladesh cord blood cohorts.

### Estimated preterm birth rate

In the Ontario cohort, the preterm birth proportion was 5.6% (5.4%, 5.8%) in the testing subset and overall based on ultrasound-determined GA. Model 3, the full model which demonstrated

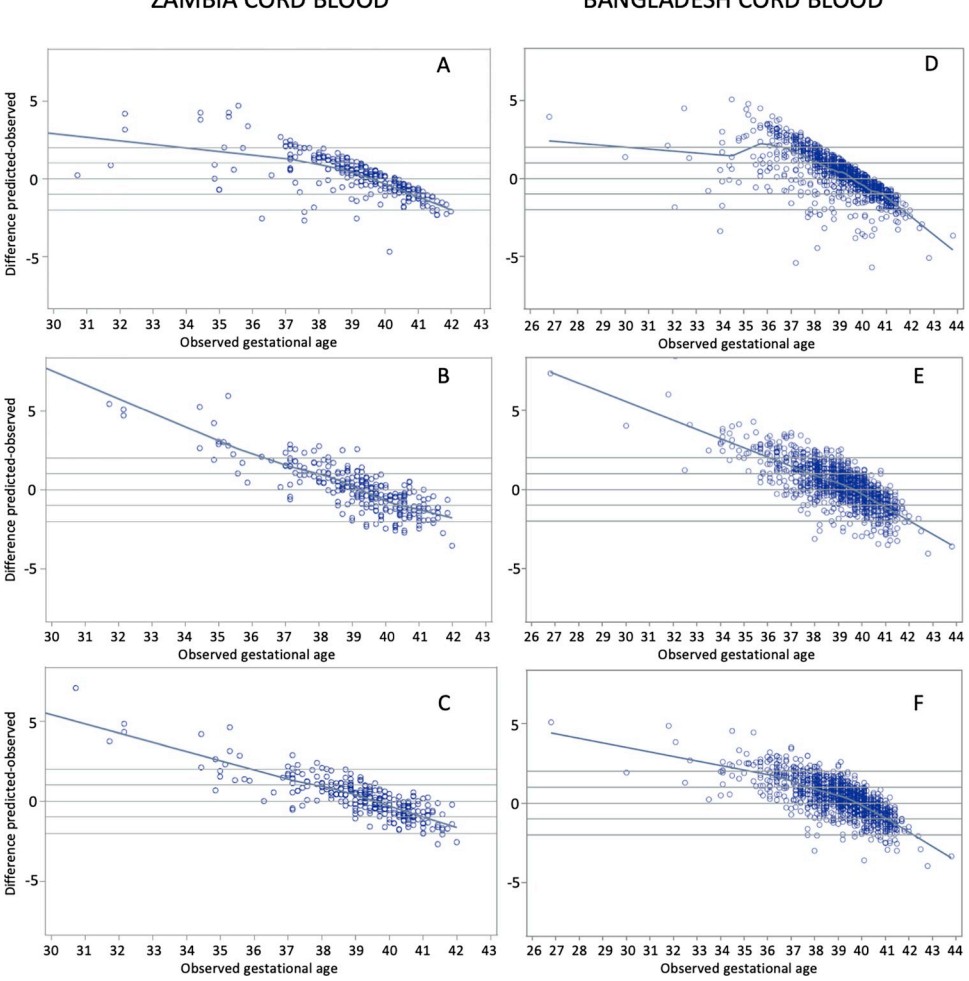

**Fig 4. Residual plots of predicted–observed by ultrasound-assigned gestational age. Cord blood samples from Zambia:** (A) Model 1: Baseline Model, (B) Model 2: Analyte Model, and (C) Model 3: Full Model. **Cord blood samples from Bangladesh:** (D) Model 1: Baseline Model, (E) Model 2: Analyte Model, and (F) Model 3: Full Model.

the highest accuracy, estimated the preterm birth proportion to be 5.7% (5.4%, 6.0%) in the Ontario testing subset. In the Zambia heel prick cohort, the observed preterm birth proportion was 7.7% (3.5%, 12.7%) and Model 3 estimated the preterm birth proportion as 5.8% (2.1%, 8.6%). In the Zambia cord blood cohort, the preterm birth proportion was 8.3% (5.1%, 11.7%) and Model 3 estimated the preterm birth proportion to be 4.4% (1.9%, 7.2%). In the Bangladesh heel prick cohort, the observed preterm birth proportion was 6.7% (4.6%, 9.0%) and Model 3 estimated the proportion as 3.8% (2.3%, 5.4%). In the Bangladesh cord blood cohort the observed preterm birth proportion was 9.5% (7.9%, 11.2%) and Model 3 yielded an estimated preterm birth proportion of 4.1% (3.0%, 5.3%).

## Classification accuracy

In the Ontario heel prick testing cohort, 1485/2226 preterm infants were correctly classified (sensitivity = 66.7%) and 36680/37440 term infants were correctly classified (specificity = 98.0%) by Model 3. In the Bangladesh heel prick cohort, 14/35 preterm infants were correctly classified as preterm (sensitivity of 40%) and 479/485 term infants were correctly classified as term (specificity of 96.8%) by Model 3. In the Zambia heel prick cohort, 6/11 preterm infants were correctly classified (sensitivity = 54.5%) and 130/131 term infants were correctly classified (specificity = 99.2%) by Model 3. Classification accuracy was markedly lower for Models 1 and 2 in all cohorts, and all 3 models were less accurate in the cord blood cohorts from Bangladesh and Zambia compared to heel prick results.

## Discussion

In this study, we demonstrated that our ML algorithms for postnatal GA estimation developed from heel prick blood sample data in Ontario, Canada, can be successfully applied in low and middle income countries in Sub-Saharan Africa and South Asia, having some of the highest preterm birth rates globally [10, 11]. When applied to heel prick samples from Lusaka, Zambia and Matlab, Bangladesh, our GA estimates from Model 3 (our best-performing model) were within an average of 6 days of ultrasound-based GA. All models produced the most accurate estimates in full term infants (37 to 39 completed weeks GA), however Model 3 provided clinically important improvements (by a week or more in some instances) in accuracy over Model 1. Although Model 1's overall performance appeared satisfactory, its precision was poor in preterm infants, particularly in growth restricted infants (SGA3 and SGA10) across the spectrum of GA. This is not surprising since Model 1 relies only on sex, birth weight and multiple versus singleton birth to estimate GA. In growth restricted infants especially, birth weight is a misleading measure of GA. Model 3 demonstrated improved accuracy compared to Model 1 in SGA infants across the Ontario, Zambia and Bangladesh cohorts, in most cases with minimal to no overlap in 95% confidence intervals. In Ontario infants, point estimates for Model 3 MAE in SGA10/SGA3 infants were 1.13/1.48 respectively, compared to 2.70/3.85 for Model 1. In Zambia, MAE for SGA10/SGA3 infants was 0.87/1.29 compared to 1.79/3.54 for Model 1. In Bangladesh the MAE for SGA10/SGA3 infants was 0.81/0.97 for Model 3 compared to 1.14/1.73 for Model 1.

We validated the performance of our GA models, which were derived using heel-prick samples, in umbilical cord blood from the Zambia and Bangladesh cohorts. Estimation of GA using cord blood would provide several advantages. It would reduce the sample collection burden on staff, results in comparably less stress for the newborn and parents and requires less training compared to heel prick sample collection.

Unfortunately, when we applied our GA models to cord blood samples we observed sharply diminished accuracy of predictions in both the Zambia and Bangladesh cohorts. There were

two main reasons for this. First, our models were derived using heel prick samples. We were unable to derive cord blood-specific models as cord blood is not routinely collected in our Ontario population. Second, our investigations in paired heel and cord samples showed that only a small subset of cord blood analyte levels are closely correlated with heel prick analytes levels in the same infants, so decreased accuracy is not surprising [26]. Hence, it may be challenging or impossible to develop a cord-blood specific model with acceptable accuracy. Nonetheless, Models 2 and 3 still demonstrated some clinical utility in estimating GA using cord blood samples.

As a sensitivity analysis, we identified analytes with high and low agreement based on Spearman correlation, and derived restricted versions of Model 3 including only analytes meeting a minimum concordance threshold. These restricted models increased the precision of cord blood GA estimates in preterm infants, but not overall (see Supplemental Results in S1 File).

In our Results, we presented scatter plots of observed versus estimated GA (Figs 3 and 4) that showed a systematic tendency for GA to be overestimated in increasingly preterm infants. This was observed for all three models in heel prick and cord blood samples from Zambia and Bangladesh. The same phenomenon was not observed in the Ontario internal validation results. There are a few possible explanations for this apparent systemic bias which can be framed as a model calibration problem. For example, our reference models employed shrinkage penalization during model fitting (via ELASTICNET regression) to control over-fitting, which may have led to over-penalized final regression model coefficients, leading to the opposite problem of under-fitting. However, this was not observed in our internal validation in Ontario data-only in the external validation cohorts. Another potential contributor is the standardization process during data preparation. The same standardization was employed in all three cohorts, but local means and standard deviations were used specific to each cohort. Local standardization was a critical step, and all models had much poorer performance when this step was omitted, however large differences in the dispersion of model predictors across populations, driven by local factors (such as socioeconomic conditions, climate, and underlying differences in birth weight and GA distributions) were likely much larger contributors to differences in covariate distributions. This could have contributed to clinically important variation being muted in the external cohorts. An important implication of this model calibration issue is that preterm (<37 weeks GA) birth estimates based on our models are too low. This is in large part due to "edge effects" which are an important limitation of dichotomizing a naturally continuous variable (for example a GA estimate of 36.9 weeks would be classified preterm while one of 37.0 would be classified as term, despite the estimates being less than a day apart). Model calibration in new external settings is an important consideration and a focus of ongoing investigation by our research team.

Direct comparisons of model performance across populations and between our current ML GA estimation models and previously reported conventional regression models, is challenging due to different distributions of GAs and birth weights, and other infant- and setting-specific factors. However, comparing overall internal validation results among previously published models and those presented in this manuscript, we noted improvements in the accuracy of GA estimates overall (RMSE = 1.06 and 1.04 vs. 0.89 for current Model 3), and in preterm infants (RMSE = 1.78 and 1.35 vs. 1.16 for current Model 3)[13, 14]. These findings were largely consistent across the internal and external validation cohorts. Therefore, our interpretation is that benefits of our ML approach were clear but represented incremental but clinically relevant improvements, suggesting that our previous models were robustly developed.

Our study had several important strengths. These include our strategy of both internally validating our models in Ontario test data, as well as our engagement with international

collaborators to externally validate in mother-infant cohorts from low- and middle-income countries. Samples were collected from infants with GA confirmed by early pregnancy ultrasound, and analyzed centrally (in the same lab where samples were analyzed for the Ontario cohort). The Zambian Preterm Birth Prevention Study (ZAPPS) and Preterm and Stillbirth Study, Matlab (PreSSMat) cohorts, in which our study was conducted, ensured that enrollment was open to representative populations of women and newborns in both Lusaka, Zambia and Matlab, Bangladesh. Other strengths include the high quality of samples received and our collection of paired heel and cord blood samples which allowed the comparison of model performance metrics between sample types, as well analyte level comparisons in paired heel prick and cord blood samples from a large subgroup of infants. The superior accuracy of Model 3 (our best performing metabolomic model) in estimating GA in SGA and preterm infants compared to Model 1 (that only relies on clinical variables) is an important strength, given the limitations reported for commonly-used tools in the postnatal period (i.e., Ballard or Dubowitz scores) in these vulnerable infants [6, 7, 27]. The primary limitation of this study is the limited number of preterm infants available in our external validation cohorts, especially very and extremely preterm infants (28–32 and <28 weeks gestation respectively). There was a higher likelihood that consent would be provided for the collection of cord blood than for heel prick samples in increasingly preterm infants. This was an important finding as it highlights the advantages of further developing estimation models that work well in cord blood samples. In the Bangladesh cohort, parents expressed reluctance to subject their premature newborns to blood sample collection (via heel prick). Although this was not systematically surveyed at the Zambia site, our study nurses reported a similar hesitancy among Zambian parents. Consequently, the relatively small number of heel prick samples collected from very preterm infants limited our ability to interpret model performance in these subgroups. Our nurses also reported a stigma surrounding the heel prick blood sample collection being perceived as associated with early infant diagnosis of HIV. Although the study team focused educational efforts on dispelling this perception, it had a persistent effect on our ability to recruit the targeted number of participants.

Implementation of our postnatal GA estimation models in LMIC settings to determine population level GA distribution was the end goal of our research, but would present many challenges. The current process involving of local sample collection and international analysis has been successful in our proof of concept, external validation and feasibility work. However, implementing the analysis pipeline locally would be ideal. Many LMIC are lacking in pathology and laboratory medicine services. Barriers include insufficient infrastructure, properly trained personnel and insufficient quality, standards and accreditation education and training for laboratory and medicine personnel [28]. Many newborn screens use tandem mass spectrometry, which has limited availability in many LMIC settings.

## Conclusion

Accurate ascertainment of preterm birth rates across LMIC is imperative in order to evaluate the impact of policies and programs aimed at improving pregnancy and neonatal outcomes. In North America, statistical models using data from biochemical analysis of newborn dried blood spots, including those previously developed by our team, have been shown to provide accurate estimates of GA, with some limitations in preterm and growth restricted infants [14–19]. In this study we have presented internal and external validation results of our most current ML algorithms employing ELASTIC NET regression for GA estimation in both high and low income settings, providing incremental improvements in performance compared to previously developed models. Large-scale implementation of this approach, and population-

level collection and analysis of newborn samples, offers a new opportunity to provide surveillance of the burden of preterm birth in jurisdictions where data are currently lacking.

## Supporting information

**S1 File. Supplemental study methods and results.**
(DOCX)

## Acknowledgments

The authors would like to thank the study participants and their families for contributing to this study.

## Author Contributions

**Conceptualization:** Steven Hawken, Kumanan Wilson.

**Data curation:** Kathryn M. Denize, Monica Lamoureux, Matthew Henderson, Katelyn J. Rittenhouse, Joan T. Price, Humphrey Mwape, Bellington Vwalika, Patrick Musonda, Jesmin Pervin, A. K. Azad Chowdhury, Anisur Rahman, Jeffrey S. A. Stringer.

**Formal analysis:** Steven Hawken, Robin Ducharme, Wei Cheng.

**Funding acquisition:** Malia S. Q. Murphy, Jeffrey S. A. Stringer, Kumanan Wilson.

**Investigation:** Pranesh Chakraborty.

**Methodology:** Steven Hawken, Robin Ducharme, Malia S. Q. Murphy, Brieanne Olibris, Wei Cheng, Julian Little, Beth K. Potter, Pranesh Chakraborty.

**Project administration:** Robin Ducharme, Malia S. Q. Murphy, A. Brianne Bota, Lindsay A. Wilson, Kathryn M. Denize, Monica Lamoureux, Humphrey Mwape, Bellington Vwalika, Patrick Musonda.

**Resources:** Matthew Henderson, Joan T. Price, Jesmin Pervin, Anisur Rahman, Pranesh Chakraborty, Jeffrey S. A. Stringer, Kumanan Wilson.

**Supervision:** Steven Hawken, Lindsay A. Wilson, Matthew Henderson, Jesmin Pervin, Pranesh Chakraborty, Kumanan Wilson.

**Validation:** Steven Hawken.

**Visualization:** Steven Hawken.

**Writing – original draft:** Steven Hawken, Robin Ducharme, Malia S. Q. Murphy, Brieanne Olibris.

**Writing – review & editing:** Steven Hawken, Robin Ducharme, Malia S. Q. Murphy, Brieanne Olibris, A. Brianne Bota, Lindsay A. Wilson, Wei Cheng, Julian Little, Beth K. Potter, Kathryn M. Denize, Monica Lamoureux, Matthew Henderson, Katelyn J. Rittenhouse, Joan T. Price, Pranesh Chakraborty, Jeffrey S. A. Stringer, Kumanan Wilson.

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
