## [Decision Letter · Decision Letter 0]

3 Aug 2022

PONE-D-22-01354

Development and external validation of machine learning algorithms for postnatal gestational age estimation using clinical data and metabolomic markers

PLOS ONE

Dear Dr. Hawken,

Thank you for submitting your manuscript to PLOS ONE. After careful consideration, we feel that it has merit but does not fully meet PLOS ONE’s publication criteria as it currently stands. Therefore, we invite you to submit a revised version of the manuscript that addresses the points raised during the review process.

We look forward to receiving your revised manuscript.

Kind regards,

Francesca Crovetto

Academic Editor

PLOS ONE

Journal Requirements:

6. Please note that in order to use the direct billing option the corresponding author must be affiliated with the chosen institute. Please either amend your manuscript to change the affiliation or corresponding author, or email us at plosone@plos.org with a request to remove this option.

7. Please amend the manuscript submission data (via Edit Submission) to include authors Kathryn M. Denize, Monica Lamoureux, Matthew Henderson and Joan T. Price.

8. Please upload a new copy of Figures 1-4 as the detail is not clear. Please follow the link for more information: https://blogs.plos.org/plos/2019/06/looking-good-tips-for-creating-your-plos-figures-graphics/" https://blogs.plos.org/plos/2019/06/looking-good-tips-for-creating-your-plos-figures-graphics/

Reviewers' comments:

Reviewer's Responses to Questions

**Comments to the Author**

1. Is the manuscript technically sound, and do the data support the conclusions?

Reviewer #1: Yes

2. Has the statistical analysis been performed appropriately and rigorously? 

Reviewer #1: I Don't Know

3. Have the authors made all data underlying the findings in their manuscript fully available?

Reviewer #1: No

4. Is the manuscript presented in an intelligible fashion and written in standard English?

Reviewer #1: Yes

5. Review Comments to the Author

Reviewer #1: Steven Hawken and colleagues presented three machine learning-based models to estimate GA shortly after birth using clinical and metabolomics data. The models were trained and internally validated with data from Ontario - Canada and externally validated with data from Zambia and Bangladesh. I found the manuscript very interesting and it is well written. However, there are some issues that the authors should address before its publication.

- In Table 3, it would be nice to add the p-value to see whether there are statistical differences between the 3 cohorts.

- There are a couple of mistakes in Table 3: In the third column, in the birgh weight Overall row it says: 31020 grams. In addition, on the row below, there is a space missing between the mean and (SD) values.

- Lines 231-232: “Accuracy of estimated GA was generally lower in the external validation cohorts than in the Ontario internal validation results for the same models” à “Accuracy of estimated GA was generally lower in the external validation cohorts than in the Ontario internal validation COHORT for the same models”

- I cannot see Tables 4 and Table 5 properly, since they are cut.

- I am unable to see the numbers on the axis of Figure 2… Can you please make the numbers bigger? Same comment applies to Figures 3 and 4.

- Lines 325 – 334: Instead of giving the percentage of preterm births estimated by each model and for each cohort, it would be more interesting to give the number of infants correctly classified as preterm by the 3 models, and for the 3 cohorts, since, as stated by the authors, one of the main objectives of their work is the properly identification of premature infants at birth, in middle and low-income countries. When do so, please provide also, other quantitative measurements such as sensitivity, specificity, etc. for preterm detection,

- Have the infants included in this study any important disease, such as, for example congenital heart disease?

- The performance of the 3 models on correctly predicting GA in preterms is quite low (Model 3: only 28.3% of the 11 preterm infants in Zambia cohort have an absolute error o equal or below to 1 week, when using heel prick samples, for example), as seen in the results provided in tables 4 and 5. Have the authors thought on ways to improve the model performance on this subset of sample? Have the authors considered the use of other sources of data together with heel trick and clinical data? What about data imputation? Can this have a negative effect on model performance when applied to preterm subset?

- Please, carefully review your Supplementary material document, since there are some characters that are not displayed properly (i.e.: “Ontario data including only the analytes in the model with a minimum Spearman correlation coefficient of 0□5 (Model 4) and 0·3 (Model 5)”. Also, the decimal points used sometimes is wrong… It should be 0.91 instead of 0·91, for example.

- Supplementary method; “We also calculated the percentage of infants with gestational ages correctly estimated within 7 and 14 days of ultrasound-based gestational age” I thought, according to the methods described in the main manuscript, that you calculated the percentage of infant with GA correctly estimated within 1 and 7 days, since this would correspond to an error of +/-1 week. However, GA estimated within 7 and 14 days would be +/-2 weeks…

6. PLOS authors have the option to publish the peer review history of their article (what does this mean?). If published, this will include your full peer review and any attached files.

Reviewer #1: No

---

## [Author Response · Author response to Decision Letter 0]

29 Sep 2022

We have uploaded a Response to Reviewer document with all comments addressed.

---

## [Editor Report · Decision Letter 1]

11 Oct 2022

PONE-D-22-01354R1Development and external validation of machine learning algorithms for postnatal gestational age estimation using clinical data and metabolomic markersPLOS ONE

Dear Dr. Hawken,

Thank you for submitting your manuscript to PLOS ONE. After careful consideration, we feel that it has merit but does not fully meet PLOS ONE’s publication criteria as it currently stands. Therefore, we invite you to submit a revised version of the manuscript that addresses the points raised during the review process.

**Reviewer #1**

Steven Hawken and colleagues presented three machine learning-based models to estimate GA shortly after birth using clinical and metabolomics data. The models were trained and internally validated with data from Ontario - Canada and externally validated with data from Zambia and Bangladesh. I found the manuscript very interesting and it is well written. However, there are some issues that the authors should address before its publication.

- In Table 3, it would be nice to add the p-value to see whether there are statistical differences between the 3 cohorts.

- There are a couple of mistakes in Table 3: In the third column, in the birgh weight Overall row it says: 31020 grams. In addition, on the row below, there is a space missing between the mean and (SD) values.

- Lines 231-232: “Accuracy of estimated GA was generally lower in the external validation cohorts than in the Ontario internal validation results for the same models” � “Accuracy of estimated GA was generally lower in the external validation cohorts than in the Ontario internal validation COHORT for the same models”

- I cannot see Tables 4 and Table 5 properly, since they are cut.

- I am unable to see the numbers on the axis of Figure 2… Can you please make the numbers bigger? Same comment applies to Figures 3 and 4.

- Lines 325 – 334: Instead of giving the percentage of preterm births estimated by each model and for each cohort, it would be more interesting to give the number of infants correctly classified as preterm by the 3 models, and for the 3 cohorts, since, as stated by the authors, one of the main objectives of their work is the properly identification of premature infants at birth, in middle and low-income countries. When do so, please provide also, other quantitative measurements such as sensitivity, specificity, etc. for preterm detection, 

- Have the infants included in this study any important disease, such as, for example congenital heart disease? 

- The performance of the 3 models on correctly predicting GA in preterms is quite low (Model 3: only 28.3% of the 11 preterm infants in Zambia cohort have an absolute error o equal or below to 1 week, when using heel prick samples, for example), as seen in the results provided in tables 4 and 5. Have the authors thought on ways to improve the model performance on this subset of sample? Have the authors considered the use of other sources of data together with heel trick and clinical data? What about data imputation? Can this have a negative effect on model performance when applied to preterm subset?

- Please, carefully review your Supplementary material document, since there are some characters that are not displayed properly (i.e.: “Ontario data including only the analytes in the model with a minimum Spearman correlation coefficient of 0□5 (Model 4) and 0·3 (Model 5)”. Also, the decimal points used sometimes is wrong… It should be 0.91 instead of 0·91, for example.

- Supplementary method; “We also calculated the percentage of infants with gestational ages correctly estimated within 7 and 14 days of ultrasound-based gestational age” I thought, according to the methods described in the main manuscript, that you calculated the percentage of infant with GA correctly estimated within 1 and 7 days, since this would correspond to an error of +/-1 week. However, GA estimated within 7 and 14 days would be +/-2 weeks…

We look forward to receiving your revised manuscript.

Kind regards,

Francesca Crovetto

Academic Editor

PLOS ONE

---

## [Author Response · Author response to Decision Letter 1]

15 Dec 2022

All responses to reviewer comments are detailed in our "Response to Reviewers" document attached to this submission.

---

## [Editor Report · Decision Letter 2]

16 Jan 2023

Development and external validation of machine learning algorithms for postnatal gestational age estimation using clinical data and metabolomic markers

PONE-D-22-01354R2

Dear Dr. Hawken,

We’re pleased to inform you that your manuscript has been judged scientifically suitable for publication and will be formally accepted for publication once it meets all outstanding technical requirements.

Kind regards,

Iman Al-Saleh

Academic Editor

PLOS ONE

Additional Editor Comments:

The authors satisfactorily addressed the reviewers' comments.

---

## [Editor Report · Acceptance letter]

24 Feb 2023

PONE-D-22-01354R2 

Development and external validation of machine learning algorithms for postnatal gestational age estimation using clinical data and metabolomic markers 

Dear Dr. Hawken:

I'm pleased to inform you that your manuscript has been deemed suitable for publication in PLOS ONE. Congratulations! Your manuscript is now with our production department. 

Kind regards, 

on behalf of

Dr. Iman Al-Saleh 

Academic Editor

PLOS ONE